# Suicide and Neurotrophin Factors: A Systematic Review of the Correlation between BDNF and GDNF and Self-Killing

**DOI:** 10.3390/healthcare11010078

**Published:** 2022-12-27

**Authors:** Stefania De Simone, Maria Antonella Bosco, Raffaele La Russa, Simona Vittorio, Nicola Di Fazio, Margherita Neri, Luigi Cipolloni, Benedetta Baldari

**Affiliations:** 1Section of Forensic Medicine, Department of Clinical and Experimental Medicine, University of Foggia, 71122 Foggia, Italy; 2Department of Anatomical, Histological, Forensic and Orthopaedic Sciences, Sapienza University of Rome, 00161 Rome, Italy; 3Department of Medical Sciences, Section of Forensic Medicine, University of Ferrara, 44121 Ferrara, Italy

**Keywords:** BDNF, GDNF, suicide, autopsy, neurotrophin, self-killing, hippocampus, prefrontal cortex, amygdala

## Abstract

According to WHO data, suicide is a public health priority. In particular, suicide is the fourth-leading cause of death in young people. Many risk factors of suicide are described, including individual-, relationship-, community-, and societal-linked ones. The leading factor is the diagnosis of mental illness. Nevertheless, not all people who attempt suicide are psychiatric patients; these characteristics help define high-risk populations. There are currently no useful biomarkers to indicate the risk of suicide. In recent years, neurotrophic factors have increasingly become of scientific interest. This review aims to summarize the current scientific knowledge on the correlation between BDNF and GDNF and suicide, to theorize whether neurotrophins could be a reliable marker for an early diagnosis of suicidal risk. The authors conducted a systematic review following PRISMA criteria. They found eight research papers in agreement with the inclusion criteria. According to the results of these studies, there may be a connection between BDNF brain levels and complete suicide, although there are discrepancies. A lack of interest in GDNF may suggest less involvement in the suicidal dynamic. Further studies may provide helpful information to researchers.

## 1. Introduction

According to WHO data, suicide is a public health priority; every year, more than 700,000 people commit suicide, and many people attempt suicide [1]. Suicide is the fourth-leading cause of death in young people (15–29 years old). There are many risk factors for suicide, including individual-, relationship-, community-, and societal-linked ones [2]. The main ones are the diagnosis of mental illness (such as major depression, anxiety, and bipolar disorders) [3], followed by drug abuse, history of early trauma, familial history of suicide attempts, and trait impulsiveness [4,5]. Nevertheless, not all people who attempt suicide are psychiatric patients; while these characteristics help define high-risk populations, they are not always relevant in personal assessment [6,7].

Suicide has a complex neurobiology that follows the diathesis–stress model, which is a framework for understanding the development of psychological disorders. According to this model, a behavior disorder results from an interaction between genetic predisposition vulnerability and stress, usually caused by life events [8,9]. Stressful life events are often triggers of suicidal behavior [10]. Psychiatric patients are at increased risk of attempting suicide [11], particularly depressed patients [12]. There are currently no useful biomarkers to indicate the risk of suicide, and even some studies performed on people who have attempted suicide have not been significant [11,12,13,14].

In recent years, neurotrophic factors have increasingly become of scientific interest. Neurotrophins are essential for the survival, maintenance, and regeneration of specific neuronal populations in fetal and adult brains [15]. The mammalian neurotrophins activate one or more of the three members of the tropomyosin-related kinase (Trk) family of receptor tyrosine kinases (TrkA, TrkB, and TrkC) [16]. All neurotrophins activate the p75 neurotrophin receptor (p75NTR), a member of the tumor necrosis factor receptor superfamily [17]. Some important neurotrophins are brain-derived neurotrophic factor (BDNF) and glial cell-derived neurotrophic factor (GDNF). Their depletion can cause several diseases.

BDNF is one of the most distributed and extensively studied mammalian neurotrophins. BDNF signals through the TrkB receptor and the low affinity p75 neurotrophin receptor (p75NTR). BDNF plays a significant role in the growth, development, and plasticity of glutamatergic and GABAergic synapses. Through modulation of neuronal differentiation, it influences serotonergic and dopaminergic neurotransmission [18]. BDNF plays a role in resilience and antidepressant drug action; it is a common genetic locus of risk for mental illnesses and is one of the most prominently psychiatric-studied molecules [19]. BDNF depletion can lead to psychiatric disorders, such as depression, anxiety [20,21], attention-deficit/hyperactivity disorder [22], and drug addiction [23]. Stress, which probably is the “lowest common denominator” risk factor for several mental illnesses, targets BDNF in disease-implicated brain regions and circuits. Dysregulation of BDNF, such as the Val66Met polymorphism, may improve susceptibility to stress, trauma, and risk of stress-induced disorders [24,25,26], highlighting the role of the epigenetic component. The BDNF Met allele (Val66Met) was associated with a history of suicide attempts [27,28], but other studies did not find a correlation [29].

Nevertheless, the majority of neurotrophin studies are in vivo, measuring the BDNF levels on plasma [11,30,31,32]. At the same time, it could be interesting to study the BDNF concentration in the brain of the victims of suicide, since not all the people who commit suicide have a diagnosis of depressive disorder or other psychiatric conditions. Postmortem studies of the brains of suicide victims revealed abnormally low levels of BDNF and its receptor, TrkB, compared to controls [33,34]. Interestingly, this was true regardless of the psychiatric diagnosis. Low BDNF protein levels were found in the amygdala [35], anterior cingulate cortex (ACC) [36], hippocampus, and prefrontal cortex [33,34].

GDNF protects catecholaminergic cells from the toxic insults expressed mainly in dopaminergic neurons [37]. Its potential therapeutic applicability in neurodegenerative diseases (such as Parkinson’s or Alzheimer’s) has been intensely investigated [38]. Some studies investigated the involvement of GDNF in mood dysregulation [39,40]. In one postmortem study, an increase in GDNF was found in the parietal cortex of patients with depressive disorder, while in others a reduction in the expression of its mRNA was found [41,42]. However, the current knowledge of GDNF’s involvement in the suicidal phenomenon is still unknown. Another study investigates the GDNF concentration in many cortical regions of depressed patients, showing a significant increase in the parietal cortex [43].

This review aims to summarize the current scientific knowledge on the correlation between BDNF and GDNF and suicide, to theorize whether neurotrophins could be a reliable marker for an early diagnosis of suicidal risk. Furthermore, this review aims to form the basis for an experimental case study.

## 2. Materials and Methods

### 2.1. Eligibility Criteria

The present systematic review followed the Preferred Reporting Items for Systematic Review (PRISMA) standards [44].

### 2.2. Search Criteria and Critical Appraisal 

We performed a systematic literature search, a critical appraisal of the collected studies, and an electronic search of Medline/PubMed and Scopus on papers published from 1 January 2015 to 1 July 2022. The search terms were: “BDNF”, “GDNF”, “brain-derived neurotrophic factor”, “glial cell-derived neurotrophic factor”, and “suicide” in the title, abstract, and keywords. Bibliographies of all identified documents were reviewed and compared for any ulteriorly relevant literature. We made a methodological evaluation of each study according to the PRISMA standards, including an assessment of bias. Data collection involved study selection and data extraction. 

### 2.3. Inclusion and Exclusion Criteria 

Inclusion criteria were research papers with original data regarding measuring BDNF and GDNF (in postmortem brain tissue, cerebrospinal fluid, plasma, serum, whole blood, and urine) and written in English. Included studies investigated the association between levels of BDNF and GDNF and suicide by comparing BDNF and GDNF levels between people committing suicide and victims of accidental traumatic death. Exclusion criteria were: non-English papers, review papers, metanalysis, communications, abstracts, articles focused on suicidal behavior, and articles about major depression.

## 3. Results

We collected 317 articles on Medline and PubMed and 224 on Scopus (541 in total), removing 163 duplicates. Of the 378 papers screened, we included only 8. We excluded 32 manuscripts because they were reviews, 8 because they were written in a foreign language, and 1 because it was only a data article [45]. The remaining papers were excluded because they disagreed with the review’s objectives (many articles focused on suicidal behavior, or the study sample included living people).

The methodology of our search strategy is shown in Figure 1.

The results of the review are summarized in Table 1 and discussed in the following paragraphs.

### 3.1. Study Characteristics

All studies concerned the deceased and analyzed brain tissues; only one study [48] used venous blood. In particular, four studies [46,50,52,53] examined the hippocampus, four [46,47,50,51] the Broadman area 10 (BA10 or prefrontal cortex), and two [49,52] the BA9, while only one study [49] focused on the BA24 and the brain stem. Three articles [48,51,52] included investigated blood, and only one [51] investigated cerebrospinal fluid (CSF). Only Gadad et al. [51] studied the GDNF; all researchers analyzed the expression of the BDNF protein; two [52,53] concerned BDNF mRNA; one study [52] investigated the methylation of its DNA; and another study [53] investigated the expression of the TrkB protein. All studies use molecular biology (Western blot, PCR, and immunoassay) or genetic (microarray) techniques.

### 3.2. Postmortem Brain

The authors analyzed a total of 198 brain samples and 149 controls. Only in four studies [46,50,52,53] was the method of suicide known, primarily by hanging (in 100 cases).

Three studies [46,49,50] measured the BDNF protein with Western blot, while Gadad et al. [51] measured it with multiplex-based Luminex analysis. Erbay et al. [53], on the other hand, did not report the values of the BDNF protein and only carried out PCR. In four studies [48,49,52,53], PCR was used to evaluate gene expression.

### 3.3. Postmortem Venous Blood

Ropret et al. [48] analyzed venous blood only, using PCR in 486 cases and 289 controls. They analyzed seven single nucleotide polymorphisms (SNPs) of the BDNF gene, selected for their involvement in psychiatric pathologies and suicidal behaviors. In the absence of blood, the authors used “formalin-fixed paraffin-embedded tissue”. The authors did not specify which tissue samples they used. Samples were analyzed by genotyping and PCR.

### 3.4. Postmortem Cerebrospinal Fluid

Only Gadad et al. [51] analyzed CSF in association with brain and plasma samples. The authors verified the connection between the concentrations of neurotrophic factors and interleukins (IL-6 and IL-1beta) in the brain, cerebrospinal fluid, and plasma with a multiplex assay.

## 4. Discussion

This systematic review aimed to evaluate and summarize the innovative literature about the association between BDNF and GDNF levels and completed suicide. The eight analyzed papers are case-control studies that regarded the BDNF protein or gene, while only one study investigated the GDNF protein. A lack of interest in GDNF may suggest less involvement in the suicidal dynamic.

Some studies have shown a significant association between reduced BDNF levels and complete suicide. According to Hayley et al. [46], Schneider et al. [47], and Erbay et al. [53], BDNF can be a promising biomarker of suicidal tendencies because peripheral levels can match with central brain levels.

Hayley et al. [46] found different BDNF protein expression in the two sexes, which is also in agreement with the epidemiological differences regarding suicidal tendencies between men and women [54]. They found that BNDF levels were lower in the prefrontal cortex (PFC) of depressed women, while in males, there was a reduction in the hippocampus. Furthermore, the basal PFC BDNF protein was lower in males than in females in non-depressed controls, which may be explained by sex differences in the epidemiology of depression and suicidal behaviors. The difference in BDNF transcription could be due to sex hormones; for example, estradiol can regulate the expression of BDNF [55].

Schneider et al. [47] compared DNA methylation of the PFC brain in suicidal males and controls through a list of candidate genes, including BDNF. Due to the small sample, the researchers found no individual genes showing significant methylation differences. However, they demonstrated greater methylation of the BDNF promoter in the frontal cortex, in agreement with other studies that found the same evidence in the Wernicke area [56]. In any case, the authors observed that a single gene cannot influence the epigenetics of suicide but that an accumulation of adverse life experiences can increase suicidal risk. Kang et al. [57] noted that BDNF methylation also increased in the blood of suicidal patients. Hence, the authors observed that the methylation patterns in the blood might reflect the methylation patterns in the brain.

Misztak et al. [50] showed a reduced level of the BDNF protein in the brains of suicide victims. The authors highlighted the role of the epigenetic component in the expression of BDNF, particularly in histones. According to other authors, the decreased levels of the BDNF protein may be related to the reduction in histone acetylation and the increase in deacetylation and methylation [58,59]. The authors pointed out that BDNF expression is tissue-dependent, particularly in the frontal cortex and hippocampus, which is consistent with other studies [60]. These epigenetic factors are further correlated with depression, suggesting an increasingly close link between depression and suicide [61].

Erbay et al. [53] quantitatively assessed the mRNAs and protein levels of BDNF, NGF, and their TrkA and TrkB receptors in the hippocampus. The authors found a significant reduction in neurotrophin proteins and receptors. They also aimed to strengthen the connection between neurotrophins and suicide and identify them as biomarkers of psychiatric pathology. A strength of their study was the sample size and homogeneity, which were greater than those of the other studies analyzed in this systematic review.

The other analyzed studies, on the other hand, did not show a significant correlation between BDNF reduction and suicide.

Indeed, Youssef et al. [49] investigated the association between the Val66Met polymorphism of the BDNF gene and low BDNF levels and suicide, major depression, and childhood adversity in brain samples. No differences between suicidal and non-suicidal subjects were shown for the polymorphism and the protein, while an association was evident with depression.

Gadad et al. [51] showed that the central (brain and CSF) and peripheral (plasma) levels of BDNF, GDNF, and IL-6 are connected, hypothesizing a dysregulation of the blood–brain barrier. In depression, the increased permeability of the membrane allows the various molecules to pass [61,62]. According to the authors, the plasmatic concentration of IL-6, BDNF, and GDNF could reflect the brain concentration. They found that the brain BDNF levels in suicide were significantly higher than in non-suicide, consistent with other studies [63,64]. Nevertheless, they showed higher BDNF levels in patients treated with medications, confirming the relationship between therapy and BDNF increasing [65].

Ropret et al. [52] examined BDNF methylation and BDNF transcriptase levels in the brain and blood. There was no difference in BDNF methylation between the two groups, while there was a reduction in the BDNF protein levels in the blood of suicide victims. They highlighted a higher expression of BDNF transcript I-IX in the Broadmann area 9 of the brain of suicide victims but not in the hippocampus.

One crucial factor is the presence of psychiatric comorbidities, such as major depression. In the research of Hayley et al., all samples were affected by major depressive disorder and had not been on antidepressant drugs for at least two months, which was confirmed by toxicological analysis. Gadad et al. [51] divided the sample based on mood disorders and/or alcohol or drug consumption, with or without psychiatric medications. It showed a consistent difference in the BDNF levels between the medication groups.

Erbay et al. [53] included only depressed suicidal subjects in the samples, excluding others with psychiatric or neurologic comorbidities and alcohol or drug users. Even in Schneider’s [47] samples, there were only depressed patients. They did not divide the samples into subgroups (based on possible confounding factors such as the use of drugs, comorbidities, etc.) because the literature [66,67,68,69] suggests that suicide-associated DNA methylation patterns are independent of the underlying psychiatric disorders. The two groups of Youssef et al. [49] did not differ in comorbidity, blood alcohol level, or other confounding factors.

According to the results of these studies, there may be a connection between BDNF brain levels and complete suicide, although there are discrepancies. The authors use different analysis methods and samples, making it difficult to draw unambiguous conclusions. Another element is the presence of the suicide method in only four studies [46,50,52,53]. Without these elements, it is challenging to hypothesize whether BDNF depletion could be related to a specific suicidal mechanism. Identifying a suicide marker remains a challenge for the scientific and forensic community; they can also be useful in the differential diagnosis in forensic pathology [70,71].

## 5. Conclusions

This systematic review’s findings partially agree with the hypothesis that a lower level of BDNF leads to an increased risk of suicide. There may be an association between central and peripheral BDNF levels, suggesting a possible use of BDNF measurement in the blood of patients at a risk of suicide. There are no specific biological markers for psychiatric pathologies. Nevertheless, the studies are few. It is necessary to conduct more in-depth studies with a standardized methodology. The most important limitation of the various studies is their sample size. This is probably due to the difficulty of carrying out autopsies in some cases of suicide and obtaining informed consent from the relatives of the deceased. More studies on non-living subjects are necessary. Few brain areas have been investigated, mostly just the hippocampus and prefrontal cortex. Further studies may provide helpful information to researchers.

In conclusion, the need for further research emerges from the study of the literature. Our group aims to deepen this research topic, so we are collecting specific samples from the brains of both suicide victims and controls to investigate them with other tools, such as immunohistochemistry, which is widely used in forensic pathology [72,73].

## Figures and Tables

**Figure 1 healthcare-11-00078-f001:**
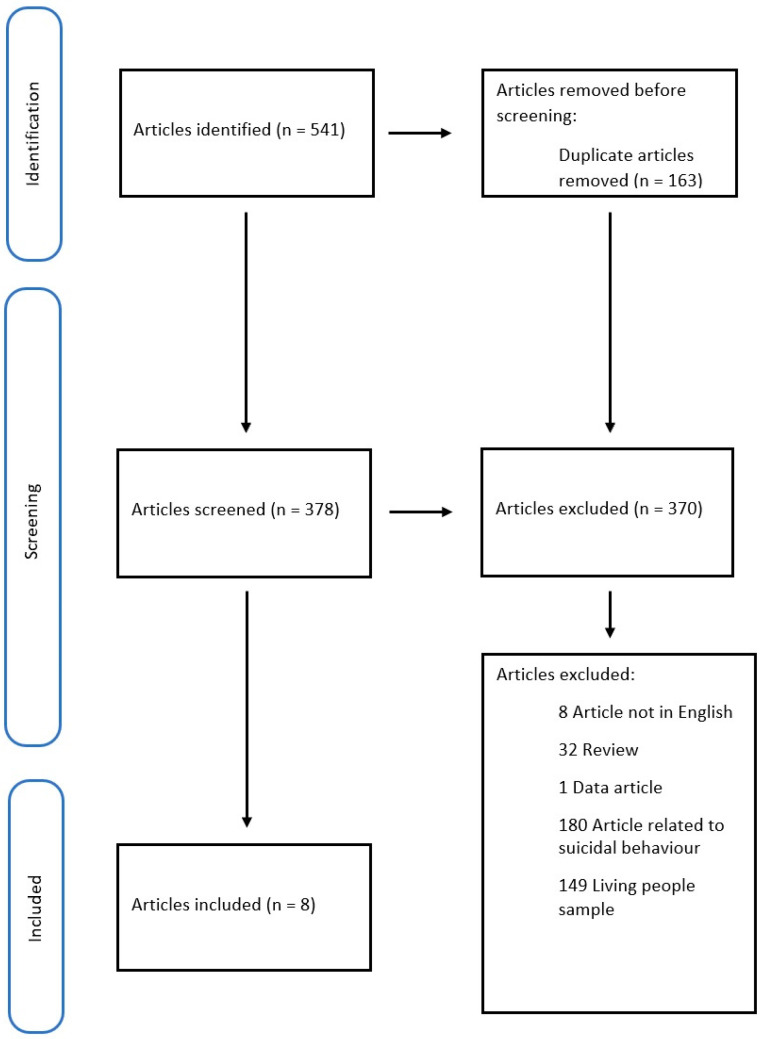
Search strategy following PRISMA criteria [44].

**Table 1 healthcare-11-00078-t001:** Characteristics of the analyzed papers.

Authors	Study Design	N. of Cases	Method of Suicide	PMI	Control	Samples	Molecule Targeted	Method of Analysis
Hayley (2015) [46]	Case-control depressed suicide individuals vs. non-suicide	19	10 hangings 8 overdoses 1 fall from height	<6 h	19	Hippocampus PFC (BA10)	BDNF protein	Western blot
Schneider (2015) [47]	Case-control suicide individuals vs. non-suicide	6	NP	<1 day	6	BA10	BDNF gene	Microarray
Ropret (2015) [48]	Case-control suicide individuals vs. non-suicide	486	NP	NP	289	Venous blood	BDNF gene	Genotyping with PCR
Youssef (2018) [49]	Case-control depressed and not depressed suicide individuals vs.-non suicide	37	NP	<1 day	53	BA9, BA24, CB, and RB	BDNF Val66Met polymorphism and BDNF protein	PCR Western blot
Mysztak (2020) [50]	Case-control suicide individuals vs. non-suicide	14	7 hangings 3 overdoses 2 falls from height 1 drowning 1 overwhelmed by train	NP	8	Hippocampus PFC	BDNF protein	Western blot
Gadad (2021) [51]	Case-control MD individuals vs. MD with AUD-SUD vs. AUD-SUD vs. non-psychiatric individuals	39	NP	<2 days	18	BA10 Plasma CSF	BDNF and GDNF protein	Multiplex-based Luminex assay
Ropret (2021) [52]	Case-control suicide individuals vs. non-suicide	22	Hanging	NP	20	Hippocampus BA9 Blood	BDNF DNA methylation and mRNA	NGS and PCR
Erbay (2021) [53]	Case-control depressed suicide individuals vs. non-suicide	61	Hanging	10 h	25	Hippocampus	BDNF, TrkB mRNA, and protein	PCR

PFC: prefrontal cortex; BA: Brodman area; NP: not present; SNPs: single nucleotide polymorphism; CB: caudal brainstem; RB: rostral brainstem; CSF: cerebrospinal fluid; PCR: polymerase chain reaction; MD: mood disorder; AUD: alcohol use disorder; SUD: substance use disorder; NGS: next-generation sequencing.

## Data Availability

Not applicable.

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
