# Peer review of "Suicide and Neurotrophin Factors: A Systematic Review of the Correlation between BDNF and GDNF and Self-Killing"

_healthcare, 2022, doi:10.3390/healthcare11010078_

Round 1

Reviewer 1 Report

General comments: this study is of interest and summarizes other studies investigating the correlation between BDNF and GDNF levels and suicide. Some general points are: 

1. Why is the period limited to only 6 months (January to July 2015)? One could wonder if it is possible to get any significant or relevant results if investigated papers are published in such a short time period (and hence only 8 papers were included in the analysis). However, I’m rather confused. The materials and methods state that the search was performed on papers published in 2015 from January 1st to July 1st, but Table 1 lists papers from 2018, 2020, and 2021 as well (references 49-53). What am I missing?

2. Table 1 very informative! Summarizes everything relevant, well done. 

Specific comments:

1. Line 33: “in young” – in young people, or young population? And would be good to define that population (what is the age interval)

2. Line 37 and Abstract: “not all suicides are psychiatric patients” – please rephrase since it sounds a bit awkward (perhaps “not all people who attempt suicide are psychiatric patients”)

3. Line 43: what factors? Vague, please rephrase

4. Line 48: Do the authors mean that the neurotrophic factors became of interest in relation to suicide research? Generally, neurotrophic factors have been investigated for a long time, so if the authors mean in relation to suicide they should say so

5. Line 72: this sentence is unclear. “…studies are in vivo on plasma” – please clarify (whose plasma? what kind of studies?) Also, line 73: concentration in the brain, not on the brain. Please rephrase the two sentences in lines 72-74.

6. Line 98: I suggest “performed” instead of “realized”

7. Line 110: only BDNF levels compared or BDNF and GDNF?

8. Line 143 (just a general note that can be discussed): how is it possible that method of suicide is not reported in all studies? It seems as important data. Maybe this fact can be mentioned briefly in the discussion.

9. Line 176: I suggest “may be explained by” instead of “consist”

Overall, the manuscript could be of interest, however investigation (analysis) on a larger scale is warranted. 

Author Response

Dear reviewer, thank you for your comments. We really appreciate your words.

1. We have corrected the text as the search period is from 2015 to 2022, we have written the sentence incorrectly.

2.Thanks!

Specific comments

We have made all the grammar corrections you indicated.

3. We removed the word "factors" as it was a typo.

4. We haven't changed the sentence as the interest is not only related to suicide, but also to the development of the brain, psychiatric pathologies and drug addiction, as we then explained in the following paragraphs. We just wanted to understand that it is a factor that has now been widely studied.

8. We have added a little comment in lines 243-245

Thanks for your precious suggestion.

Reviewer 2 Report

As the authors mentioned, BDNF can be targeted in various nonspecific conditions including stress, depression, and anxiety. Therefore, BDNF is not likely a good biological marker for suicide. Besides, there are various reasons behind each suicide cases, it is not likely to find one specific biological marker for such complicated condition. Instead, scoring from multiple factors may be helpful in identifying suicidal risks.

Author Response

Dear Reviewer, thanks for your opinion. We appreciate your point of view and your comment.

Round 2

Reviewer 2 Report

As I mentioned before, BDNF can be targeted in various nonspecific conditions including stress, depression, and anxiety. Therefore, BDNF is not likely a good biological marker for suicide. The authors need strong arguments to support their interest in linking BDNF to suicide.

Author Response

As we said in the article, and as you say, BDNF is involved in some conditions (such as depression, anxiety, and stress) related to an increased risk of suicide. For this reason, we analyzed the literature for data on BDNF concentrations in suicidal subjects. Our article tries precisely to demonstrate, by arguing and analyzing other studies, what you say. Our study is an attempt to search for new markers for suicide risk. It is an innovative review because there are no others in the literature, and it tries to find the strong evidence you mention. This probably will be the preliminary review to further studies by our research group.